# Study on Active Components of *Cuscuta chinensis* Promoting Neural Stem Cells Proliferation: Bioassay-Guided Fractionation

**DOI:** 10.3390/molecules26216634

**Published:** 2021-11-02

**Authors:** Hanze Wang, Xiaomeng Hou, Bingqi Li, Yang Yang, Qiang Li, Yinchu Si

**Affiliations:** 1School of Pharmaceutical Sciences, Changchun University of Chinese Medicine, Changchun 130117, China; wanghanze_123@163.com; 2School of Chinese Materia Medica, Beijing University of Chinese Medicine, Beijing 102488, China; houxm1224@163.com (X.H.); niclaus@163.com (B.L.); yang123yang0928@163.com (Y.Y.); 3Department of Anatomy, School of Traditional Chinese Medicine, Beijing University of Chinese Medicine, Beijing 102488, China

**Keywords:** neural stem cells, *Cuscuta chinensis*, active components, network pharmacology

## Abstract

Neural stem cells (NSCs) exist in the central nervous system of adult animals and capable of self-replication. NSCs have two basic functions, namely the proliferation ability and the potential for multi-directional differentiation. In this study, based on the bioassay-guided fractionation, we aim to screen active components in *Cuscuta chinensis* to promote the proliferation of NSCs. CCK-8 assays were used as an active detection method to track the active components. On the basis of isolating active fraction and monomer compounds, the structures of these were identified by LC-MS and (^1^H, ^13^C) NMR. Moreover, active components were verified by pharmacodynamics and network pharmacology. The system solvent extraction method combined with the traditional isolation method were used to ensure that the fraction TSZE-EA-G6 of *Cuscuta chinensis* exhibited the highest activity. Seven chemical components were identified from the TSZE-EA-G6 fraction by UPLC-QE-Orbitrap-MS technology, which were 4-*O*-p-coumarinic acid, chlorogenic acid, 5-*O*-p-coumarinic acid, hyperoside, astragalin, isochlorogenic acid C, and quercetin-3-*O*-galactose-7-*O*-glucoside. Using different chromatographic techniques, five compounds were isolated in TSZE-EA-G6 and identified as kaempferol, kaempferol-3-*O*-glucoside (astragalin), quercetin-3-*O*-galactoside (hyperoside), chlorogenic acid, and sucrose. The activity study of these five compounds showed that the proliferation rate of kaempferol had the highest effects; at a certain concentration (25 μg/mL, 3.12 μg/mL), the proliferation rate could reach 87.44% and 59.59%, respectively. Furthermore, research results using network pharmacology techniques verified that kaempferol had an activity of promoting NSCs proliferation and the activity of flavonoid aglycones might be greater than that of flavonoid glycosides. In conclusion, this research shows that kaempferol is the active component in *Cuscuta chinensis* to promote the proliferation of NSCs.

## 1. Introduction

Ischemic stroke (IS), an acute cerebral blood circulation disorder, often occurs in the elderly. In particular, cerebrovascular diseases result in a large increase in the rate of disability and mortality of diabetes [1]. In recent years, due to most IS patients unable to effectively receive treatment in the acute phase, they may inevitably enter the stroke sequelae stage. Stroke sequelae mainly refers to a condition left after hemorrhage of acute cerebrovascular disease, hemiplegia, numbness, skewed eyes, and poor speech [2,3]. Clinically, the treatment of stroke sequelae focuses on rehabilitation, lacking effective treatment methods. In recent years, some reports have shown that traditional Chinese medicine (TCM) plays an important role in the treatment of stroke sequelae [4,5]. In the case of stroke sequelae patients, it is necessary to find effective natural products in TCM to promote nerve recovery.

Neural stem cells (NSCs) are self-renewing multipotent cells, which can generate neurons, astrocytes, and oligodendrocytes. NSCs have the ability of proliferation and multi-directional differentiation [6]. Among them, the ability of proliferation is the first step to exert neural function. During the period of cerebral ischemia, NSCs in the subgranular zone (SGZ) and subventricular zone (SVZ) are activated, resulting in proliferation, migration, and differentiation function. However, the number of NSCs and newborn neurons in SGZ and SVZ is extremely limited, and the related regulatory factors are also insufficient, which require the effects from TCM [7].

According to the theory of TCM, the deficiency of kidney qi is an important pathogenesis of nerve injury diseases. The brain marrow depends on the essence of the kidney and the sea of marrow so that the relevant functions of the brain can be brought into normal play [8]. Besides that, a great number of reports have shown that kidney essence in TCM and NSCs have the same source, similar function, and distribution in the human body [9]. However, clinically, there is still a lack of relevant medicine to promote the proliferation of NSCs. Through preliminary research, we found that kidney-tonifying traditional Chinses medicine *Velvet Antler*, *Cuscuta chinensis*, *Rehmannia glutinosa*, *Achyranthes bidentata*, *Lycium barbarum*, *Morinda officinalis*, *Cistanche deserticola*, *Polygonatum sibiricum*, *Epimedii Folium*, *Cornus officinalis*, and *Polygoni Multiflori* had the activity of promoting NSCs proliferation, among which *Velvet Antler* showed the highest activity, followed by *Cuscuta chinensis*. In this study, we choose *Cuscuta chinensis* for the further research.

The seeds of *C. chinensis* (*Cuscutae Semen*), recorded in the famous book Shen Nong’s Herbal and other Chinese traditional pharmacopoeias, have become one of the most common medicine to nourish the kidney and liver [10]. *Cuscuta chinensis* is a type of parasitic plant among various species and possesses pharmacological effects on reproductive system, bone metabolism, anti-aging, anti-inflammatory, immune system, etc. *Cuscuta chinensis* has many effects on the function of the reproductive system, which can promote hypothalamic–pituitary–gonadotropic axis (HPGA) function, improve pituitary responsiveness to gonadotropin-releasing hormone, and follicular development [11,12]. According to the TCM theory, the kidney and brain are interrelated in function, the imbalance of HPGA is the fundamental and key to the occurrence of “disharmony between kidney and brain”. However, although *Cuscuta chinensis* has a variety of activities for nourishing kidney, there have been no reports with regard to active components promoting the proliferation of NSCs in *Cuscuta chinensis* until recently.

The objective of this study was to screen active components in *Cuscuta chinensis* to promote the proliferation of NSCs, which contribute to the discovery of related drugs with clinical application potential.

## 2. Results

### 2.1. Morphological Observation of NSCs

The primary NSCs isolated from the cerebral hemisphere of fetal rats were in the state of small round dots under the microscope (1 × 100), evenly dispersed and bright in the center (Figure 1A).

After 48 h, most of the cells gathering into clusters with strong refraction, and some black spots appeared in the visual field. Due to the culture medium of NSCs was a special culture medium for serum-free NSCs, black spots might represent the death of miscellaneous cells or NSCs in poor conditions (Figure 1B).

After 72 h, the cells were clustered in a large area to form hundreds of neurospheres with strong refraction, consistent shape, but with different sizes (Figure 1C).

After 4–5 days, a small number of single cells could be seen under the microscope (1 × 200), the neurosphere was suspended in a large area, so it needed to be subcultured as soon as possible in this state (Figure 1D).

After subculturing, the neurospheres were dispersed into single cells. Due to the removal of miscellaneous cells and dead cells, the visual field became clearer (Figure 1E).

### 2.2. Activity Screening of Different Fraction of Cuscuta Chinensis

According to Table 1 and Figure 2, TSZE-DCM and TSZE-EA have the highest proliferation rate (51.87% and 53.31%) at a concentration of 3.12 μg/mL and 1.56 μg/mL.

In conclusion, the active components in TSZE-DCM and TSZE-EA of *Cuscuta chinensis* have strong proliferation-promoting activity of NSCs. In this study, the TSZE-EA part of *Cuscuta chinensis* with strong activity was selected for the next step in order to obtain the relevant active components.

### 2.3. Activity Screening of TSZE-EA from Different Isolated Parts

According to Table 2 and Figure 3, all the fractions of TSZE-EA except EA-G1 had the activity of promoting the proliferation of NSCs. Among them, EA-G6 at different concentrations (25 μg/mL, 12.5 μg/mL, 6.25 μg/mL, 3.12 μg/mL, 1.56 μg/mL) all showed strong proliferative activity. The proliferation rate had reached 76.57% at a concentration of 12.5 μg/mL, which was significantly different from the control group (*p* < 0.001).

In summary, most of the isolated parts of the EA extract of *Cuscuta chinensis* had the activity of promoting the proliferation of NSCs, in which the TSZE-EA-G6 fraction showed the highest cell proliferation activity at different concentrations. Therefore, the active components of the TSZE-EA-G6 fraction needed to be further clarified.

### 2.4. Identification of Chemical Components in TSZE-EA-G6 Fraction by UPLC-QE-Orbitrap-MS Technology

The TSZE-EA-G6 fraction dissolved in methanol and had a constant volume concentration of 1 mg/mL. The solution passed through a 0.22 pm miscroporous membrane to obtain the test solution, which was tested on the Agilent 1100 high-performance liquid chromatograph to explore the isolation condition. In order to detect the compound results more comprehensively, two detection modes, positive and negative, were used (Figure 4).

According to Table 3, the components of the TSZE-EA-G6 fraction were mainly flavonoids and quinic acid compounds [13]. However, in order to clarify the active components promoting NSCs proliferation, the monomer compounds should be isolated.

### 2.5. Study on the Activity of Compound of Promote the Proliferation of NSCs

The experimental grouping, plating, and calculation formula were equivalent to 2.2, and the research results of compounds A to E promoting the proliferation of NSCs were as follows.

According to Table 4 and Figure 5, the experimental results showed that kaempferol at different concentrations (25 μg/mL, 12.5 μg/mL, 6.25 μg/mL, 3.12 μg/mL, 1.56 μg/mL) had the activity of promoting the proliferation of NSCs. The highest proliferation rate was 87.44% at the concentration of 25 μg/mL, which was significantly different from that of the control group (*p* < 0.001); the proliferation rate of astragalin was 13.2% at a concentration of 12.5 μg/mL; the proliferation rate of hyperoside at a concentration of 25 μg/mL was 13.37%; the proliferation rate of chlorogenic acid at a concentration of 6.25 μg/mL was 22.59%, which was significantly different from the control group (*p* < 0.05); and sucrose did not have the proliferation activity of NSCs at any concentration.

In summary, kaempferol showed the highest activity of promoting the proliferation of NSCs.

### 2.6. Explore the Active Components in the Proliferation of Cuscuta Chinensis NSCs Based on Network Pharmacology Technology

The “active-components-targets” network was established for 10 representative components in *Cuscuta chinensis*, as shown in Figure 6. Three hundred targets related to neuroprotection were found through related databases and the literature, and a disease-target network was established by Cytoscape (3.8.0) software. Among them, the targets related to neuroprotection and cell proliferation were included, non-functional targets were removed, and a network of 300 nodes and 885 nodes was obtained.

Combining the “active-components-targets” network and the “disease-targets” network to obtain the intersection targets of these two networks, including the targets AKT1, PTGS2, TNF, HMOX1, CYP1A1, PTGS1, ACHE, and CA4.

Among them, AKT1 (Serine/threonine-protein kinase 1), namely serine-threonine-protein kinase, has a degree value of 32. Since the degree value was the largest, it played the largest role in the network, as shown in Table 5.

According to the results of GO enrichment analysis, as shown in Table 6, in the biological process, items related to neuroprotection mainly included the response to oxidative stress, the positive regulation of smooth muscle cell proliferation, cellular response to hypoxia, and the regulation of blood pressure, cell proliferation, inflammation response, etc.; in the process of molecular function, it mainly included heme binding, enzyme binding, hydrogen peroxide activity, etc.; at the cellular component, it mainly included organelle membrane and endoplasmic reticulum membrane, etc.

The PI3K/AKT (phosphatidylinositol-3 kinase/protein kinase B) signaling pathway is mainly composed of phosphatidylinositol PI3K and its downstream molecule is protein kinase B Akt. AKT1 is the key kinase of this pathway, which plays a role in NSCs proliferation [14].

AKT1 is a common target of kaempferol, quercetin, and isorhamnetin in *Cuscuta chinensis*, all of which are flavonoid aglycones. After analyzing the network pharmacology prediction results and combining with the activity verification experiment, it was found that the flavonoid aglycones exhibited a profound biological effect in promoting the proliferation of NSCs, but the activity of flavonoid glycosides was not remarkable.

Among them, flavonoids exert a wide range of biological functions on several cell types and affect the fate of NSCs. In this study, for the first time, we observed the activity of kaempferol promoting the proliferation of NSCs. Kaempferol can be widely found in a variety of Chinese herbal medicines, which possess antioxidant, anticancer, and nerve protection activities [15]. Until now, research on the neuroprotective effects of kaempferol has made some progress. Kaempferol has the effect of brain-derived neurotrophic factor-like (BDNF-like) function, which can enhance the function of neurons, so as to realize brain repair after injury [15,16]. In addition, kaempferol, a type of flavonoid aglycone compound with a small molecular weight, can overcome the blood–brain barrier (BBB) from entering the brain and plays a role in the lesion, which should also be one of the reasons for its remarkably proliferative activity compared with flavonoid glycosides [16].

NSCs have the potential capacity to multi-directionally differentiate, namely being able to differentiate into neurons, astrocytes, and oligodendrocytes, and can self-renew and produce a great number of cerebral cells [6]. In recent years, natural products havepossessed certain activity in NSCs proliferation, such as ginsenoside Rg1, ligustrazine, ginkgolide and, etc. [17]. In this research, we clarified that the natural product Kaempferol showed remarkable activity, which can penetrate the BBB and played a role in neuroprotective effects. Compared with other reported flavonoids compounds, the cell proliferation rate of Kaempferol is higher than that of the reported compounds with this activity, e.g., the NSCs proliferation rate of icariin showed 12.69% (1 × 10^−6^ mol/L) [18], which lays the foundation for research related to mechanisms and druggability.

## 3. Materials and Methods

### 3.1. Chemicals and Reagents

Fetal bovine serum (FBS), DMEM/F12, basic fibroblast growth factor, penicillin-strepto-mycin liquid and trypsin were obtained from Invitrogen; the Cell Counting Kit-8 (CCK-8) was obtained from Beyotime (Beijing, China); silica gel and polyamide were obtained from E. Merck, Darmstadt, Germany; UPLC-QE-Orbitrap-MS (Thermo Fisher Scientific, San Jose, CA, USA) operated using Xcalibur 2.2 software (Thermo Fisher, San Jose, CA, USA).; semi-preparative HPLC (H&E, Beijing) with a YMC ODS H-80 or L-80 column (YMC, Tokyo, Japan) was used for the final purification. The NMR spectra were acquired on AV-500 instruments (Bruker, Switzerland).

### 3.2. Animals

A total of 6 pregnant female Sprague-Dawley rats of SPF grade were purchased from Sibeifu Experimental Animal Science and Technology Co., Ltd. (Beijing, China; no. SCXK2020-0010) on embryonic day 17 (E17). All animals were housed individually at 22 ± 2 °C and a relative humidity of 50 ± 10% with a 12 h light/12 h dark cycle. Food and water were given ad libitum throughout the experiment. All procedures in the present study were performed in accordance with the institutional guidelines and ethics of Beijing University of Chinese Medicine (Beijing, China). All surgical procedures were performed under anesthesia and all efforts were made to minimize suffering

### 3.3. Isolation and Culture of NSCs

The E17 were weighed and anesthetized by intramuscular injection of 10% chloral hydrate at the dose of 0.4 mL/100 g. The fetuses were removed from the female rats in advanced pregnancy individually following anesthetization, then the fetuses were immediately decapitated. On a sterile workbench, the meninges surrounding the brain were peeled off. The cortexes were then cut into small pieces (1 mm) in Dulbecco’s Modified Eagle’s Medium (DMEM)/F12 (1:1) after removing any remaining blood with a dissecting solution (D-Hanks, composed of KCl, KH_2_PO_4_, NaCl, NaHCO_3_, and Na_2_HPO_4_*12H_2_O). These small pieces were digested with 0.05% trypsin for 10 min. Then, an equal amount of 10% fetal bovine serum was added to terminate digestion and the tissue was washed with DMEM/F12 medium. After passing through a 200 mesh cell sieve to make a single cell suspension, the cell density of the suspension was adjusted to 1 × 10^6^/mL with serum-free DMEM/F12 culture medium supplemented with B27. These single cells were cultured in a constant temperature incubator at 37 °C and 5% CO_2_, that were, the primary NSCs [19].

After 5–7 days of culture, the spheroidization of NSCs was observed under the microscope, which were mechanically dissociated into individual cells. The cell density was adjusted to 1 × 10^6^/mL and cells were continuously cultured in a constant temperature incubator at 37 °C and 5% CO_2_.

### 3.4. Preparation of Isolated Parts of Cuscuta Chinensis

Previous research found that both the alcohol extract (TSZE) and water extract (TSZW) of *Cuscuta chinensis* showed the ability to promote the proliferation of NSCs. Among them, TSZE had a more significant proliferation ability (Table 7), therefore, TSZE was selected for further study.

*Cuscuta chinensis* seeds (500 g) were extracted twice, 1.5 h each time, via heating and ethanol refluxing. After recovering the solvent, we obtained TSZE (yield about 5.6%). After water dispersion, different polar extracts were extracted from the TSZE: petroleum ether extract (TSZE-PE, yield about 0.29%), dichloromethane extract (TSZE-DCM, yield about 0.27%), ethyl acetate extract (TSZE-EA, yield about 0.31%), n-butanol extract (TSZE-nBu, yield about 3.8%) and remaining water extract (TSZE -WA, yield about 1.12%) were obtained, respectively (Figure 7).

A fraction of the TSZE-EA extract (1.55 g) was subjected to a silica gel chromatographic column eluted with dichloromethane-methanol (1:0, 30:1, 20:1, 10:1, 5:1, 1:1, 0:1) as the mobile phase. The column volume of eluent was collected three times for each concentration and concentrated under reduced pressure to obtain seven fractions, which were Ea-G1 (yield 0.02%), Ea-G2 (yield 0.05%), Ea-G3 (yield 0.12%), Ea-G4 (yield 0.12%) Ea-G5 (yield 0.06%), Ea-G6 (yield 0.05%), Ea-G7 (yield 0.06%) (Figure 8).

### 3.5. Cell Counting Kit-8 (CCK-8) Method

Experimental group: treatment group (100 μL NSCs and 100 μL extract samples of different concentrations), medicine zero group (100 μL samples at different concentrations and 100 μL cell medium), control group (100 μL NSCs and 100 μL cell medium), and medium group (200 μL cell medium), medicine zero group and medium group were not inoculated with cells. Six samples were set in the treatment group and the control group, and three samples were set in the medicine zero group and the cell medium group.

Planking: The single-cell suspension was adjusted to the density of 5 × 10^5^/mL and inoculated into 96-well plates. In this study, three batches of cells were used to repeat the operation three times.

Administration: In the treatment group, different concentrations of extracts were added to each well, and then 96-well plates were cultured at 37 °C and 5% CO_2_ for 72 h and then detected by the CCK-8 method. After culturing for 72 h, 20 μL of CCK-8 test solution was added to each hole and mixed. After incubation for 1.5 h, the microplate reader detected the OD values at 450 nm and calculated cell proliferation rate. SPSS 25.0 statistical software was used for data processing, and the results were expressed as the mean ± standard deviation, *p* < 0.05 was considered to indicate a statistically significant difference.
$$\mathrm{cell}\;\mathrm{proliferation}\;\mathrm{rate}\;(\%)=[\frac{\mathrm{OD}\;\mathrm{treatment}\;\mathrm{group}\;-\;\mathrm{OD}\;\mathrm{medicine}\;\mathrm{zero}\;\mathrm{group}}{\mathrm{OD}\;\mathrm{control}\;\mathrm{group}\;-\;\mathrm{OD}\;\mathrm{medium}\;\mathrm{group}}-1]\;\times \;100\%$$

Data are expressed as the mean ± standard deviation.

### 3.6. UPLC-QE-Orbitrap-MS Technology

#### 3.6.1. Chromatographic Separation Conditions

Acetonitrile 0.1% formic acid water (25:75); detection time: 17 min; injection volume: 10 μL; column temperature: 25 °C; detection wavelength: 270 nm; flow rate: 0.3 mL/min.

#### 3.6.2. Mass Separation Conditions

Ion source: ESI; detection mode: positive and negative ion mode; capillary voltage: 3.800 V; atomizer pressure: 40 psi; dryer flow rate: 6 L/min^−1^; dryer temperature: 32 °C, scanning mass range: *m*/*z* 50~1500.

### 3.7. Extraction, Isolation and Screening of Active Components

*Cuscuta chinensis* seeds (5 kg) were extracted twice with 95% ethanol reflux for 1.5 h each time. After filtration, the solvent was recovered and extracted three times with the same amount of dichloromethane and ethyl acetate. After concentration, we obtained the ethyl acetate fraction (TSZE-EA) with a yield of about 0.31%.

A fraction of the TSZE-EA extract was subjected to a silica gel chromatographic column eluted with dichloromethane-methanol (1:0, 30:1, 20:1, 10:1, 5:1, 1:1, 0:1) as the mobile phase. Collected three times the column volume of eluent for each concentration and concentrated under reduced pressure to obtain six fractions, followed by TSZE-EA-G1 (yield 0.02%), TSZE-EA-G2 (yield 0.05%), TSZE-EA-G3 (yield 0.12%), TSZE-EA-G4 (yield 0.12%), TSZE-EA-G5 (yield 0.06%), TSZE-EA-G6 (yield 0.05%).

The TSZE-EA-G6 fraction was eluted with the dichloromethane-methanol system (30:1~1:10), 6 column volumes for each concentration. A total of five fractions (A1~A5) were obtained after TLC inspection (A1~A5): A1 was purified by semi-preparative HPLC to obtain compound A (50 mg); after isolation with silica gel column chromatography and polyamide column chromatography, compound B (100 mg) and C (100 mg) were obtained in A2; A3 was purified by semi-preparative HPLC to obtain compound D (50 mg); A4 was concentrated under reduced pressure and sucking filtration, we obtain compound E (100 mg).

#### 3.7.1. Kaempferol

Yellow, amorphous powder; ^1^H-NMR (500 MHz, MeOH) *δ* ppm: 6.18 (1H, d, *J* = 2 Hz, H-6), 6.39 (1H, d, *J* = 2 Hz, H-8), 8.09 (2H, d, *J* = 8.9 Hz, C2′-C6′), 6.90 (2H, d, *J* = 8.9 Hz, C3′-C5′); ^13^C-NMR (250 M Hz, MeOH) *δ* ppm: 145.4 (C-2), 135.0 (C-3), 174.7 (C-4), 159.9 (C-5), 97.00 (C-6), 162.9 (C-7), 91.8 (C-8), 155.0 (C-9), 101.9 (C-10), 121.1 (C-1′), 128.0 (C2′-C6′), 113.6 (C3′-C5′), 159.9 (C-4′).The above data is basically consistent with the data of kaempferol (C_15_H_10_O_6_) in the literature [20], so compound A is identified as kaempferol.

#### 3.7.2. Asragalin (Kaempferol 3-*O*-β-D-Glucoside)

Yellow, amorphous powder;^1^H-NMR (500 MHz, DMSO-*d*_6_) *δ* ppm: 12.64 (1H, s, H-5), 6.21 (1H, d, *J* = 2 Hz, H-6), 10.89 (1H, s, H-7), 6.44 (1H, d, *J* = 2 Hz, H-8), 8.05 (2H, d, *J* = 8.9 Hz, H2′-H6′), 6.89 (2H, d, *J* = 8.9 Hz, H3′-H5′), 10.21 (1H, s, H-4′), 5.46 (1H, d, *J* = 7.4 Hz, H-1″), 3.17 (1H, s, H-2″), 3.21 (1H, s, H-3″), 3.07 (1H, H-4″), 3.08 (1H, s, H-5″), 3.32, 3.56 (2H, s, H-6″); ^13^C-NMR (250 M Hz, DMSO-*d*_6_) *δ* ppm: 156.8 (C-2), 134.2 (C-3), 178.4 (C-4), 161.7 (C-5), 99.1 (C-6), 164.6 (C-7), 94.1 (C-8), 156.7 (C-9), 106.1 (C-10), 121.4 (C-1′), 131.4 (C2′-C6′), 115.6 (C3′-C5′), 160.4 (C-4′), 101.2 (C-1″), 74.2 (C-2″), 76.9 (C-3″), 70.3 (C-4″), 78.0 (C-5″), 61.8 (C-6″). The above data is basically consistent with the data of Asragalin (C_21_H_20_O_11_) in the literature [20], so compound B is identified as Asragalin (Kaempferol 3-*O*-β-D-glucoside).

#### 3.7.3. Hyperoside (Quercetin-3-*O*-β-D-Galactoside)

Yellow, amorphous powder; compound C: ^1^H-NMR (500MHz, DMSO-*d*_6_) *δ* ppm: 12.69 (1H, s, H-5), 6.20 (1H, d, *J* = 2 Hz, H-6), 10.88 (1H, s, H-7), 6.41 (1H, d, *J* = 2 Hz, H-8), 7.53 (1H, dd, *J* = 8.4 Hz, J = 2 Hz, H-2′), 9.76 (1H, s, H-4′), 6.82 (1H, dd, *J* = 8.4 Hz, H-5′), 7.68 (1H, dd, *J* = 2 Hz, H-6′), 5.39 (1H, d, *J* = 7.7, H-1″), 3.56 (1H, s, H-2″), 3.36 (1H, s, H-3″), 3.64 (1H, s, H-4″), 3.31 (1H, s, H-5″), 3.29, 3.44 (2H, s, H-6″); ^13^C-NMR (250 M Hz, DMSO-*d*_6_) *δ* ppm: 156.8 (C-2), 133.9 (C-3), 178.0 (C-4)), 162.1 (C-5), 98.9 (C-6), 164.6 (C-7), 94.0 (C-8), 156.7 (C-9), 104.4 (C-10), 121.5 (C-1′), 117.1 (C-2′), 145.7 (C-3′), 148.9 (C-4′), 115.5 (C-5′), 121.5 (C-6′), 101.9 (C-1″), 72.1 (C-2″), 73.6 (C-3″), 69.1 (C-4″), 76.7 (C-5″), 60.0 (C-6″). The above data is basically consistent with the data of hyperoside (C_21_H_20_O_12_) in the literature [20], so compound C is identified as hyperoside (quercetin-3-*O*-β-D-galactoside).

#### 3.7.4. Chlorogenic Acid

Light yellow, amorphous powder; compound D: ^1^H-NMR (500 MHz, DMSO-*d*_6_) *δ* ppm: 12.43 (1H, s, -COOH), 12.60 (1H, H-1), 7.03 (1H, d, *J* = 2.4 Hz, H-2), 6.77 (1H, d, *J* = 8.4, H-5), 7.00 (1H, dd, *J* = 8.4 Hz, 2.4 Hz, H-6), 7.42 (1H, d, *J* = 16 Hz, H-7), 6.15 (1H, d, *J* = 16 Hz, H-8), 2.47 (1H, m, H-2), 3.57 (1H, m, H-3), 5.05 (1H, t, H-4′), 3.92 (1H, m, H-5′), 2.00 (1H, m, H-6′); ^13^C-NMR (250 M Hz, DMSO-*d*_6_) *δ* ppm: 175.4 (-COOH), 126.0 (C-1), 114.7 (C-2), 146.0 (C-3), 149.4 (C-4), 115.2 (C-5), 122.3 (C-6), 146.0 (C-7), 116.2 (C-8), 166.2 (C-9), 70.7 (C-1′), 36.6 (C-2′), 68.8 (C-3′), 73.8 (C-4′), 77.0 (C-5′), 37.6 (C-6′). The above data is basically consistent with the data of chlorogenic acid (C_16_H_18_O_9_) in the literature [21], so compound D is identified as chlorogenic acid.

#### 3.7.5. Sucrose

White lump crystal: ^1^H-NMR (500 MHz, DMSO-*d*_6_) *δ* ppm: 5.02 (1H, d, *J* = 5 Hz, H-1), 3.56 (1H, m, H-2), 3.78 (1H, m, H-3), 3.48 (1H, m, H-4), 3.86–3.90 (1H, m, H-5), 3.86–3.90 (2H, m, H-6), 3.65 (1H, s, H-1′), 4.78 (1H, d, *J* = 8.4, H-3′), 4.04 (1H, m, H-4′), 3.90 (1H, m, H-5′), 3.86–3.90 (2H, m, H-6′); ^13^C-NMR (250 M Hz, DMSO-*d*_6_) *δ* ppm: 92.2 (C-1), 72.1 (C-2), 73.3 (C-3), 70.3 (C-4), 73.3 (C-5), 60.9 (C-6), 62.5 (C-1′), 104.9 (C-2′), 77.4 (C-3′), 74.7 (C-4′), 83.0 (C-5′), 60.9 (C-6′). The above data is basically consistent with the data of sucrose (C_12_H_22_O_₁₁_) in the literature [22], so compound E is identified as sucrose.

On the basis of isolating the active components, we screened the activity of promoting the proliferation of NSCs.

### 3.8. Network Pharmacology Technology

Network pharmacology is an emerging approach that emphasizes the concept of “network target, multicomponent therapeutics” [23]. It highlights the mechanism of active compounds against disease by establishing the network of “compound-disease-target”. In this part, on the basis of clarifying the activity of kaempferol, further verification was carried out via network pharmacology.

Combined with activity study and related literature reports, the active components in *Cuscuta chinensis* were obtained, and the active components’ targets were searched through the Stitch (http://stitch.embl.de/, accessed on 2 February 2021) database, TCMSP (Traditional Chinese Medicine Systems Pharmacology Database) database. Furthermore, the “active-components-targets” network was mapped via Cytoscape (3.8.0) software.

Search through the OMIM (Online Mendelian Inheritance in Man, http://www.omim.org, accessed on 18 February 2021), GAD (Genetic Association Database,http://genetic association db.nih.gov, accessed on 18 February 2021), TTD (http://bidd.nus.edu.sg/group/cjttd/TTD_HOME.asp, accessed on 18 February 2021) with the keywords of “ischemic stroke” and “cerebral stroke”. At the same time, through literature search, relevant targets that can regulate the nervous system, neuroprotection, and promote cell proliferation were obtained. In addition, the database of String (https://string-db.org/, accessed on 20 March 2021) was used to clarify the interaction between disease targets, and Cytoscape (3.8.0) software was used to map the “disease-targets” network.

After combining the “active components-targets” network and “disease-targets” network via Cytoscape (3.8.0) software, we obtained the intersection target of chemical component and disease for analysis. And the obtained intersection targets were analyzed by GO (Gene Ontology) enrichment analysis using DAVID (https://david.ncifcrf.gov/, V6.8, accessed on 16 April 2021) database.

### 3.9. Statistical Analysis

All data were processed with the use of SPSS 20.0 (IBM SPSS, Armonk, NY, USA). The data are expressed as the mean ± standard deviation. The significance of variables was determined using a one-way analysis of variance (ANOVA). Compared with the control group, *p* < 0.05 was considered to indicate a statistically significant difference.

## 4. Conclusions

In this study, we used the method of activity tracking to screen the active components of *Cuscuta chinensis* to promote the proliferation of NSCs. After activity screening, the active fraction TSZE-EA-G6 was prepared. Using the UPLC-QE-Orbitrap-MS analysis method, seven compounds were identified in the fraction, namely hyperoside, astragalin, isochlorogenic acid C, chlorogenic acid, quercetin-3-*O*-galactose-7-*O*-glucoside, 4-*O*-p-coumarinic acid, and 5-*O*-p-coumarinic acid; and five monomer compounds were isolated from the TSZE-EA-G6 fraction of *Cuscuta chinensis*, namely: kaempferol, astragalin, hyperoside, chlorogenic acid, and sucrose.

After screening for activity, the results showed that kaempferol played a key role in NSCs proliferation for *Cuscuta chinensis* and the activity of flavonoid aglycone in TCM showed greater significance. Further research on this subject should focus mainly on several aspects: searching for active components in the dichloromethane part; studying the activity and structure–activity relationship of flavonoids; studying the related mechanisms of action and druggability of kaempferol.

## Figures and Tables

**Figure 1 molecules-26-06634-f001:**
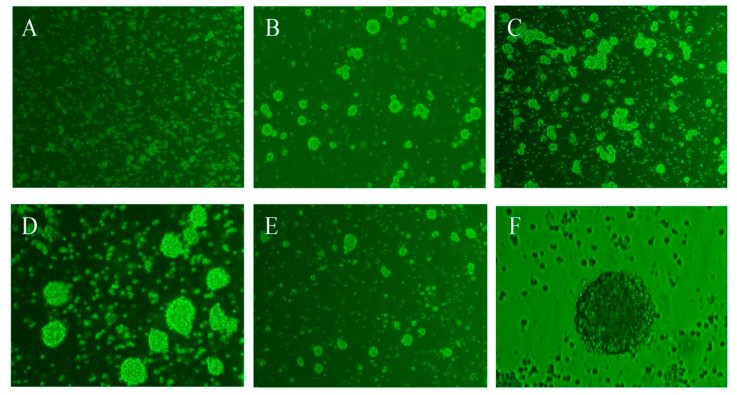
Growth state of NSCs. (**A**) Primary NSCs (100×); (**B**) NSCs cultured for 48 h (100×); (**C**) NSCs cultured for 72 h (200×); (**D**) NSCs needed to be subcultured (200×); (**E**) NSCs after subculturing (100×); (**F**) Neurospheres (400×).

**Figure 2 molecules-26-06634-f002:**
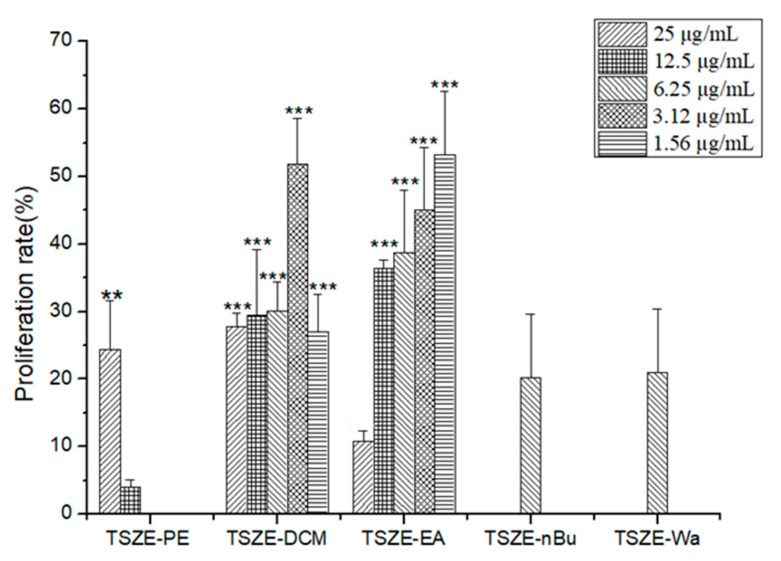
Activity of Different fraction of Cuscuta Chinensis in promoting NSCs proliferation. Different from control, ** *p* < 0.01, *** *p* < 0.001.

**Figure 3 molecules-26-06634-f003:**
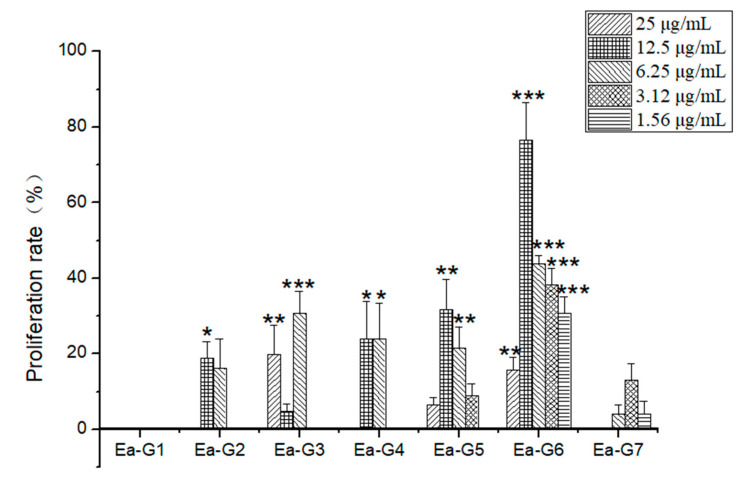
The effect of each fraction in the EA extract to promote the proliferation of NSCs. Different from control: * *p* < 0.05, ** *p* < 0.01, *** *p* < 0.001.

**Figure 4 molecules-26-06634-f004:**
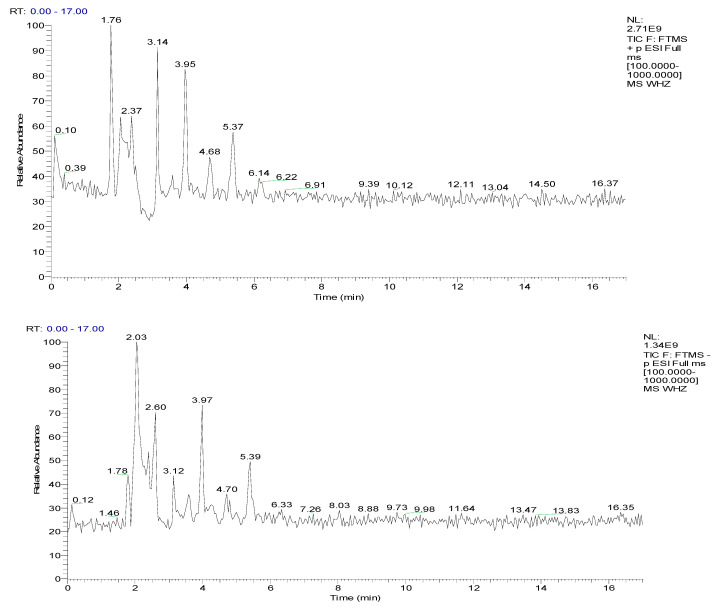
LC/MS positive and negative ion BPC diagram of TSZE-EA-G6 fraction.

**Figure 5 molecules-26-06634-f005:**
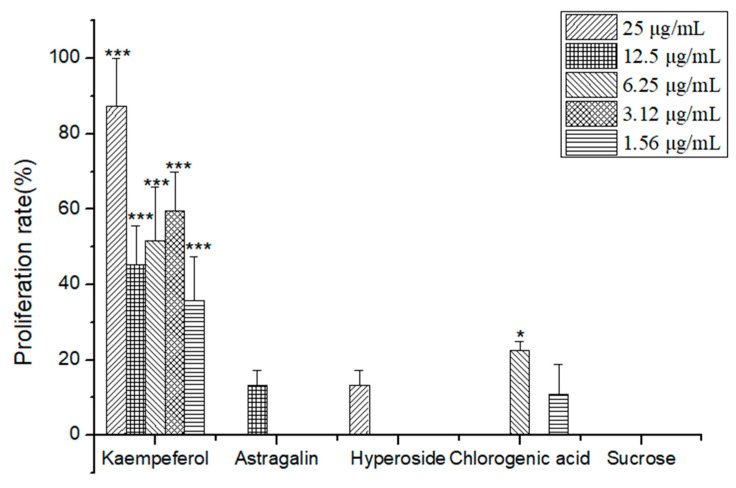
The effects of active compound in TSZE-EA-G6 on the proliferation of NSCs. Different from control, * *p* < 0.05, *** *p* < 0.001.

**Figure 6 molecules-26-06634-f006:**
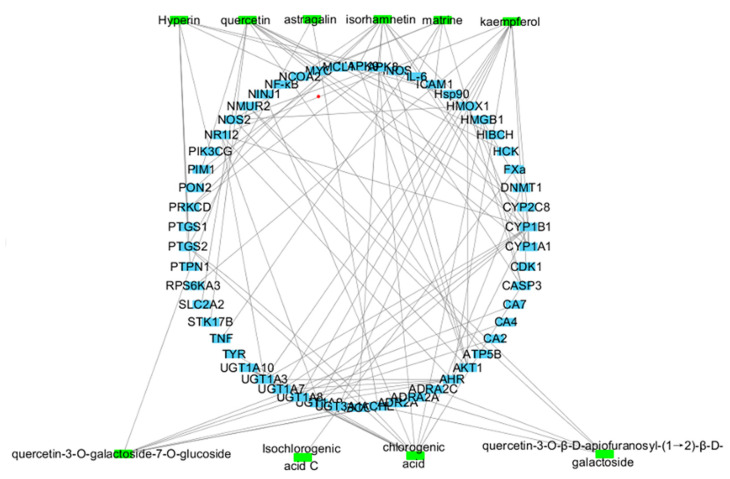
Active compounds–targets network. Green represents the active components; blue represents the targets of the components.

**Figure 7 molecules-26-06634-f007:**
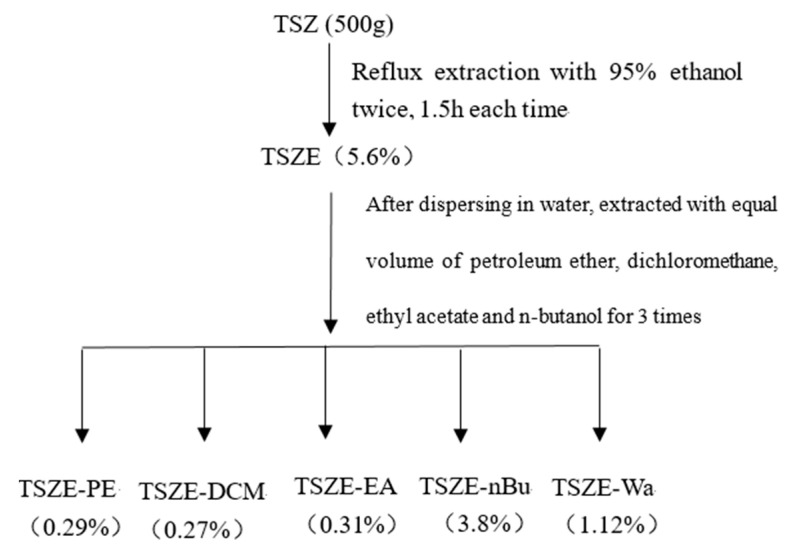
Isolation process of each part of *Cuscuta chinensis*.

**Figure 8 molecules-26-06634-f008:**
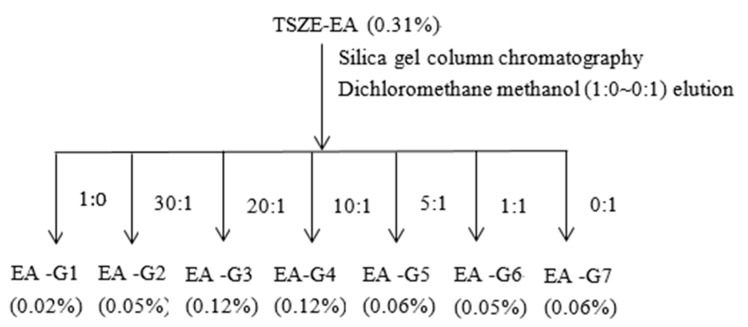
Preparation process of elution components in TSZE-EA.

**Table 1 molecules-26-06634-t001:** Proliferation of NSCs at each isolated part of *Cuscuta chinensis*.

C.	Proliferation Rate (%)				
(μg/mL)	TSZE-PE	TSZE-DCM	TSZE-EA	TSZE-nBu	TSZE-Wa
25	24.32 ± 7.21 **	27.84 ± 1.89 ***	10.87 ± 1.45	-	-
12.5	4.07 ± 1.02	29.44 ± 9.81 ***	36.48 ± 1.11 ***	-	-
6.25	-	30.18 ± 4.18 ***	38.75 ± 9.31 ***	20.30 ± 6.38	21.05 ± 7.72
3.12	-	51.87 ± 6.72 ***	44.99 ± 9.30 ***	-	-
1.56	-	27.02 ± 5.59 ***	53.31 ± 9.41 ***	-	-

Different from control, ** *p* < 0.01, *** *p* < 0.001.

**Table 2 molecules-26-06634-t002:** Effects of each fraction in the TSZE-EA to promote the proliferation of NSCs.

C	Proliferation Rate (%)						
(μg/mL)	Ea-G1	Ea-G2	Ea-G3	Ea-G4	Ea-G5	Ea-G6	Ea-G7
25	-	-	19.85 ± 7.86 **	-	6.52 ± 2.01	15.73 ± 3.35 **	-
12.5	-	18.82 ± 4.37 *	4.70 ± 2.1	24.07 ± 9.89 *	31.81 ± 8.01 **	76.57 ± 10.16 ***	-
6.25	-	16.12 ± 7.88	30.76 ± 5.72 ***	24.07 ± 9.41 *	21.53 ± 5.69 **	43.95 ± 2.12 ***	4.18 ± 2.21
3.12	-	-	-	-	8.85 ± 3.21	38.17 ± 4.42 ***	13.04 ± 2.65
1.56	-	-	-	-	-	30.87 ± 4.29 ***	4.18 ± 3.33

Different from control, * *p* < 0.05, ** *p* < 0.01, *** *p* < 0.001.

**Table 3 molecules-26-06634-t003:** LC/MS data of TSZE-EA-G6 fraction.

No.	tR/min	Chemical Formula	MS (*m/z*)	MS/MS (*m/z*)	Error	Active Compound	Structure Type
1	1.78	-	288 (−)	74.90, 158.98, 146.96, 130.98	-	-	
2	2.03	-	341 (−)	179.06, 119.04, 89.02	-	-	
3	2.57	C_16_H_18_O_8_	337 (−) 339 (+)	191.06, 173.05, 163.04 147.04	2.56 2.22	cis-4-pCoQA	Quinic Acid
4	3.12	C_16_H_18_O_9_	353 (−)	191.06, 173.05, 135.05, 93	3.26	Chlorogenic acid	Quinic Acid
5	3.61	C_16_H_18_O_8_	337 (−) 339 (+)	191.06, 173.05, 163.04 147.04	2.22 2.32	cis-5-pCoQA	Quinic Acid
6	3.97	-	283 (−)	239.08, 221.07, 195.09, 193.08	-	-	
7.	4.23	C_21_H_19_O_12_	463 (−) 465 (+)	300, 301 303.05	3.11 3.76	Hyperoside	Flavonoids
8	4.70	-	723 (−)	677.50	-	-	
9	5.39	C_21_H_20_O_1__1_	447 (−) 449 (+)	284.03, 285.03 287	3.18 1.54	Asragalin	Flavonoids
10	6.33	C_2__5_H_2__4_O_1__2_	515 (−) 517 (+)	353.09, 173.05, 191.06, 135.05, 163.04	3.37 3.26	Isochlorogenic acid C	Flavonoids
11	6.78	C_2__7_H_30_O_17_	625 (−)	463.09, 300.03, 301.04	4.23	Quercetin-3-*O*-galactose-7-*O*-glucoside	Flavonoids

**Table 4 molecules-26-06634-t004:** The effects of related monomers in TSZE-EA-G6 on the proliferation of NSCs.

C	Prolifetarion Rate (%)				
(μg/mL)	Kampferol	Astragalin	Hyperoside	Chlorogenic Acid	Sucrose
25	87.44 ± 12.6 ***	-	13.37 ± 4.0	-	-
12.5	45.3 ± 10.25 ***	13.2 ± 4.0	-	-	-
6.25	51.61 ± 14.2 ***	-	-	22.59 ± 2.3 *	-
3.12	59.59 ± 10.2 ***	-	-	-	-
1.56	35.77 ± 11.7 ***	-	-	10.84 ± 8.0	-

Different from control: * *p* < 0.05, *** *p* < 0.001.

**Table 5 molecules-26-06634-t005:** The degree value of the intersection target.

No.	Target Name	Degree
1	AKT1	32
2	PTGS2	14
3	TNF	10
4	HMOX1	9
5	CYP1A1	7
6	PTGS1	7
7	ACHE	4
8	CA4	2

**Table 6 molecules-26-06634-t006:** The results of GO enrichment analysis.

Type	Item	*p*-Value
Biological Process	positive regulation of smooth muscle cell proliferation	1.5 × 10^−6^
	response to oxidative stress	9.4 × 10^−6^
	positive regulation of apoptotic process	1.9 × 10^−4^
	regulation of blood pressure	3.1 × 10^−4^
	inflammatory response	3.7 × 10^−4^
	cellular response to hypoxia	3.6 × 10^−3^
	positive regulation of NF-kappaB import into nucleus	8.7× 10^−3^
	cell proliferation	9.3 × 10^−3^
Biological Process	heme binding	1.8 × 10^−5^
	enzyme binding	2.5 × 10^−4^
	prostaglandin-endoperoxide synthase activity	8.3 × 10^−4^
	peroxidase activity	9.1 × 10^−3^
	Protein homodimerization activity	3.4 × 10^−2^
Cellular Component	Organelle membrane	4.7 × 10^−4^
	Endoplasmic reticulum membrane	3.2 × 10^−3^
	Cell surface	1.7 × 10^−2^
	Perinuclear region of cytoplasm	2.2 × 10^−2^
	caveola	2.5× 10^−2^

**Table 7 molecules-26-06634-t007:** Proliferation activity of TSZE and TSZW on NSCs.

C	Proliferation Rate (%)	
(μg/mL)	TSZE	TSZW
100	13.96 ± 2.18 **	12.80 ± 2.12 **
50	22.04 ± 1.55 **	20.85 ± 3.03 **
25	31.04 ± 1.82 **	25.83 ± 2.16 **
12.5	20.85 ± 1.71 **	19.67 ± 1.69 **
6.25	13.98 ± 1.71 **	13.51 ± 2.79 **

Data are expressed as the mean ± standard deviation; different from control, ** *p* < 0.01.

## Data Availability

The data presented in this study are not available.

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
