# Peer review of "Study on Active Components of *Cuscuta chinensis* Promoting Neural Stem Cells Proliferation: Bioassay-Guided Fractionation"

_molecules, 2021, doi:10.3390/molecules26216634_

Round 1
Reviewer 1 Report
- Reformate the manuscript according to the journal style.
- The introduction needs to be expanded.
- At the end of the introduction, add a paragraph about the study objectives and hypothesis.
- Extensive English editing is necessary.
- The discussion is very poor.
- A conclusion must be added.
Author Response
Response to Reviewer 1 Comments
Thank you for your honest suggestions. The specific changes as following:
Point 1:The introduction was expanded. At the end of the introduction, we added a paragraph about the study objectives .
Point 2 : We made some adjustments to the format of the article.
Point 3: In the discussion, we made some comparison with some literature. And cite more references.
Reviewer 2 Report
In this submitted manuscript titled “Study on active components of Semen Cuscuta promoting neural stem cell proliferation: bio-assay-guided fractionation”, Wang et al employed CCK-8 cell proliferation assay to guide their fractionation of a traditional Chinese medicine Semen Cuscuta. As a result, the authors found that G6 fraction of the acetyl acetate extract exhibited the most potent activity to promote NSC proliferation. Subsequently, the authors identified seven known chemical components by MS/MS, isolated, and characterized five other components, amongst which kaemferol displayed the best proliferative activity. At last, through network pharmacology and gene ontology, the authors predicted that kaemferol may regulate the PI3K/Akt pathway thus elevate NSC proliferation.
Unfortunately, I found this submitted manuscript is poor in novelty and the result is inconclusive. At this stage, I regretfully do not recommend its publication by Molecules.
Major points:
1) the submitted manuscript engaged traditional activity-guided fractionation to investigate the active ingredient of TCM, although the authors isolated or identified several chemical constituents in Semen Cuscuta by tandem mass or via chromatography, none of the structures is new.
2) the authors although showed kaemferol displaying the best activity to promote NSC proliferation, no other substantial evidence other than prediction was provided to support the connection between kaemferol and PI3K/Akt pathway, for example, western blot.
Other comments:
- In the introduction, there is no sufficient description of why the authors specifically investigate Semen Cuscuta, what pharmacognosy research has been done for this species?
- There are related literature reports mentioned in several places throughout the manuscript, I highly recommend the authors cite those indispensable articles in the Reference.
- There are figures or tables not cited in the manuscript, for example, figure 4, figure 6, figure 7, figure 8, table 5…
- The authors showed the same result in both table and figure format in the manuscript, for example, table 2 and figure 3, table 3 and figure 5…, this seems redundant.
- How many biological replicates were performed for the cell proliferation assay?
- Error bar should be added to all figures.
- What tests were performed in all the statistics?
- How did the author get 50-100mg of compounds A-E? my calculation: 5kg Semen Cuscuta can give 5000g*0.31%*0.05%=0.00775g of Ea-G6.
Author Response
Response to Reviewer 2 Comments
Thank you for your honest suggestions. The specific changes as following:
Point 1: The objective of this study was to screen active components in Cuscuta chinensis to promote the proliferaton of NSCs, which contribute to discover related drugs with clinical application potential. So, in this research, we pay more attention to screening active compounds.
Point 2: The emphasis of this research is the activity of compound. Network pharmacology is used to verify the activity of kaempferol .And there have been some reports shown that kaempferol has the effect of brain derived neurotrophic factor-like (BDNF-like) function and can overcome the blood-brain barrier (BBB).
Point 3:In the introduction of this article, we carried out the activity comparative study of Cuscuta chinensis and other kidney tonifying Chinese medicines. Cuscuta chinensis seeds showed remarkable activity of promoting the proliferation of NSCs.
Point 4 : In terms of details, we supplement the citation of relevant references and error bar in figures.
Point 5 :In the article of 3.5 part,we cited supplement “three batches of cells were used to repeat the operation three times.”
Point 6 :The significance of variables was determined using one-way analysis of variance (ANOVA).(3.9)
Point7: In the article of 3.7 part, 5kg The specific changes as following: Cuscuta chinensis can give 5000g*0.05%=2.5g of Ea-G6. Ea-G6 was 0.05% of seeds.
Reviewer 3 Report
Please check the use of capitol letter, in the abstract.
In the introduction section, please check the name of the plants. Additionally, the description of the aims of the study could be more descriptive.
Regarding the experimental procedure, the number of animals, and the statistical to calculate it, should be indicated.
The organization chart of the manuscript is unusual. According to the journal formatting rules, it should: "Abstract", "Introduction", "Results", "Discussion", "Materials and Methods", "Conclusions". It is confusing to have "Methods and results". Authors should correct this aspect.
The statistical analysis paragraph is missing, and it is not clear which statistical test was used.
The network pharmacology approach is also too briefly discussed. Although authors used the cytoscape software to create the targets-components analysis, they did not state the bioinformatics platforms used for such evaluations.
The discussion section is also lacking about comparison with literature.
Regarding the number of references, eight references are really a few.
It is also to highlight that only one reference is included in the introduction section.
The improvement of the comparison with literature has to be a crucial aspect during the revision of the manuscript.
Author Response
Response to Reviewer 3 Comments
Thank you for your honest suggestions. The specific changes as following:
Point 1 : The introduction was expanded. At the end of the introduction, we added a paragraph about the study objectives.
Point 2 ; In the article , we cited the experimental animals and the method of Statistical Analysis (3,2 、3.9 )
Point 3:We made some adjustments to the format of the article.
Point 4 :In the discussion, we made some comparison to some literature and cited more references.
Point 5 :In the part of network pharmacology,we clarify the reason for apply this technology(3.8).
Round 2
Reviewer 1 Report
The manuscript is now OK.
Reviewer 2 Report
In this submitted revision titled “Study on active components of Cuscuta chinensis promoting neural stem cell proliferation: bioassay-guided fractionation”, Wang et al has kindly addressed most of the questions that I raised with the first submission.
I recommend its publication in Molecules, however, prior to that, the authors are recommended to further address the following minor points:
minor comments:
- There are still couple of figures not cited in the manuscript, e.g., figure 7, 8…
- Please correct typos: kampferol in table 4 should be kaempferol
- 13C-NMR should be 125 MHz
- For chemicals, numbers should be in subscript format, e.g., NaHCO3
Reviewer 3 Report
Manuscript has been significantly improved after revision.
Some minor changes need to finalize the correction:
-the plant name has to be in italics in all manuscript sections;
-the name of kaempferol is not correct in table 4.